# Transcriptional Modulation of the Host Immunity Mediated by Cytokines and Transcriptional Factors in *Plasmodium falciparum*-Infected Patients of North-East India

**DOI:** 10.3390/biom9100600

**Published:** 2019-10-11

**Authors:** Md Zohaib Ahmed, Nitin Bhardwaj, Supriya Sharma, Veena Pande, Anupkumar R Anvikar

**Affiliations:** 1Epidemiology and Clinical Research Division, ICMR-National Institute of Malaria Research, New Delhi 110077, India; zohaiba01@gmail.com (M.Z.A.); nitinbh529@gmail.com (N.B.); supsmicro@gmail.com (S.S.); 2Department of Biotechnology, Kumaun University, Bhimtal, Uttarakhand, 263136, India; veena_kumaun@yahoo.co.in

**Keywords:** malaria, cytokines, transcriptional factors, innate immunity, cell-mediated immunity, north-east India

## Abstract

Complications due to malaria are caused mostly by host immunological responses. *Plasmodium falciparum* subverts host immunity by various strategies, including modulation in the host immune responses by regulating cytokines. The transcriptional alterations of major cytokines and immunoregulators were analyzed in this study through gene expression profiling in clinically defined subgroups of *P. falciparum* patients. Malaria patients were included from Dhalai district hospital of Tripura with uncomplicated malaria (UC) and severe malaria (SM) and healthy controls from endemic and non-endemic areas of India. qPCR gene expression analysis was performed for all factors and they were grouped into three clusters based on their altered expressions. The first cluster was downregulated with an increased parasitic burden which included T-BET, GATA3, EOMES, TGF-β, STAT4, STAT6 and cytokines IFN-γ, IL-12, IL-4, IL-5, and IL-13. RANTES, IL-8, CCR8, and CXCR3 were decreased in the SM group. The second cluster was upregulated with severity and included TNF-α, IL-10, IL-1β and IL-7. PD-1 and BCL6 were increased in the SM group. The third cluster comprised of NF-κB and was not altered. The level of perforin was suppressed while GrB expression was elevated in SM. *P. falciparum* malaria burden is characterized by the modulation of host immunity via compromization of T cell-mediated responses and suppression of innate immune-regulators.

## 1. Introduction

Malaria is a multifactorial, life-threatening disease [1] and continues to be a public health problem worldwide with approximately 219 million infected cases worldwide in 2017 by the World Health Organization [2]. *P. falciparum,* which is known to cause severe malaria, is responsible for about half the malaria burden in India [3]. *P. falciparum* malaria is widespread in rural areas of the country. However, endemicity is stratified differently for different parts of the country. North-eastern states (Arunachal Pradesh, Assam, Manipur, Meghalaya, Mizoram, Nagaland, Sikkim, and Tripura) of the country account for only 4% of the country’s population, yet contribute to 10% *P. falciparum* cases [4]. Tripura, the study state, has reported 12,390 malaria cases in 2018, most of them being *P. falciparum* [3]. These north-eastern states are prone sites as the first case of chloroquine drug resistance *P. falciparum* malaria in India was too documented in Assam in the year 1973 [5]. These areas continue to be at risk of drug resistance, with reduced efficacy of artesunate + sulpha-pyrimethamine in *P. falciparum* [6]. This resistance pressure is thought to infiltrate through the international borders along these states [4,7] The immunological responses of the host in the area may be altered due to this drug resistance pressure. 

*P. falciparum* malaria is prone to become severe due to various factors like PfEMP1, cytoadherence or sequestration, rosetting, etc. [8]. Apart from the parasitic factors, host immunity plays a role in deciding the disease outcome. Coordination of pro- and anti-inflammatory cytokines network is well known to contribute to disease progression and have diverse functionality which could be accountable for either causing malaria severity or disease resolution. They differentiate between the diverse forms of clinical malaria through differential regulation of T_H_1 and T_H_2 paradigm [8,9,10,11]. Although, previous studies have been reported to comprehend the regulations of cytokines and associated molecules in malaria [8,11,12], a detailed study on transcriptional profiling of host immune responses by cytokines and other associated signaling molecules in malaria is much needed. Their precise involvement in terms of different clusters and groups should be described in detail to conclude how the parasitic factors are regulating host’s molecular signaling during malaria progression. For an improved illustration of the role displayed by CD4 + T_H_1 and T_H_2 together with factors of innate immunoregulators like CD8+Tc cells, macrophage, and natural killer [NK] cells in malaria pathogenesis, transcriptional profiling of associated chief cytokines and other regulatory molecules were analyzed in the present study in diverse clinical conditions. The signaling proteins selected in the panel consisted of T_H_1 specific T-BET (TBX21), STAT1 (signal transducer and activator of transcription), STAT4, lymphotoxin alpha (LT-α), TNF-α, IFN-γ and IL-12Rβ2. T_H_2 specific GATA3, STAT6, IL-4, IL-5, IL-13, and IL-10. Some general transcription factors which are critical in immunological signaling included were NF-κB, cMAF, AP-1(cJUN), NFAT1, BATF, RUNX1, and RUNX3. Signaling proteins for innate immune responses selected were CSF1 (macrophage), IL-7, IRF1, ETS1, PERFORIN (PERF), GRANZYME-B (GrB) and EOMES. Chemokine associated molecules like CCR8, CXCR3, RANTES (CCL5), and CXCL8 (IL-8) along with significant factors like TGF-β, IL-1β, IL-12 and p38MAPK were also included. The expression of negative regulators of transcription like SOCS1 (suppressor of cytokine signaling), SOCS3, BCL6 and PD1 were measured. 

All the transcription factors, cytokines, and chemokines which were included in the study, suggested a compromised T_H_1 and T_H_2 status and distressed innate immunoregulation. Chemokines and its receptors along with the other major transcription factors were found to be downregulated during the malaria progression. Though, some of the critical immunoregulators were found to be elevated which may endow with some protection against *P. falciparum* infection significantly in severe malaria.

## 2. Materials and Methods

### 2.1. Study Site and Population

The study was carried out at Dhalai district hospital, Dhalai, Tripura, India. Tripura shares its major parts with the international border of Bangladesh. Dhalai is the largest district among the total eight districts of Tripura and is an endemic site for *P. falciparum* malaria in north-east India. The district’s 70% area comprises dense forests and hills with an annual rainfall of about ~2–2.5 m [4]. The area has a hot and humid climate with an average temperature range from 17 to 36 °C and relative humidity of about 70–80%, thus making it highly suitable for malaria transmission. Ethnic tribes staying in remote hilly areas predominate the district. The district is categorized as the most socio-economically backward with poor healthcare services and minimal disease and prevention awareness, making it difficult for malaria control program operations [13].

### 2.2. Study Design

A total of twenty-two *P. falciparum* malaria-infected and nine healthy control subjects were enrolled for the study from September 2016 to October 2017. Malaria was diagnosed using microscopy and confirmed malaria patients were treated according to the national drug policy [14]. Parasite density was estimated by Giemsa-stained thick and thin blood smear examination in a conventional light microscope. Patients were categorized into uncomplicated and severe malaria groups on the basis of severity criteria classified according to the World Health Organization [15]. Other than severe malaria (SM; *n* = 05), uncomplicated cases were stratified into uncomplicated malaria1 (UC1; *n* = 05) and uncomplicated malaria2 (UC2; *n* = 12) based on parasite density where UC1 < 25,000 parasite/μL and UC2 > 25,000 parasite/μL, respectively. Healthy endemic control samples (EC; *n* = 06) were collected from patient’s relatives with no malaria infection. Samples of healthy non-endemic controls (NEC; *n* = 03) with no malaria history were also collected from other non-endemic areas (Bihar state) for comparison and analysis purposes.

### 2.3. Sample Collection

Peripheral venous blood samples (1 mL) were withdrawn into Tempus blood RNA tube (Applied Biosystems, Foster City, CA, USA) from the enrolled *P. falciparum* malaria patients and healthy controls. It was mixed several times gently and stored at −80 °C until use. Other clinical parameters were obtained and mentioned in demographic details (Table 1).

### 2.4. Sample Preparation and cDNA Synthesis

Before processing, samples were kept at room temperature for a minimum of two hours. Total RNA was extracted using the Tempus RNA isolation kit (Applied Biosystems, Foster City, CA, USA) following the manufacturer’s instructions. The isolated RNA was quantified using NanoDrop spectrophotometer (Thermo Scientific, Waltham, MA, USA). cDNA synthesis was carried out with 500 ng RNA as starting material by high capacity cDNA reverse transcription kit (Applied Biosystems, Foster City, CA, USA) using random hexamer primer.

### 2.5. Primer Designing and Gene Expression Analysis

RT-qPCR primers were designed using Primer3 software as per method defined elsewhere [16]. Desalted primers were synthesized by GCC Biotech (India) Pvt. Ltd. (Kirtankhola, West Bengal, India). Accuracy and specificity of primers were checked by Blast algorithm and their products were electrophoretically checked on 1.5% agarose gel for amplification accuracy. Accession numbers were obtained from the DNA Data Bank of Japan (DDBJ) for all primers (Table 2). cDNA was diluted in a ratio of 1:5 by adding molecular grade water. Real-time PCR was performed on LightCycler 480, Roche, in 20 μL reaction volumes in 96-well plate (Bio-rad, Hercules, CA, USA) containing 2 X SYBR green PCR master mix (KAPA Biosystems, Wilmington, MA, USA), 0.25μM of each primer, cDNA and water to adjust the volume using standard conditions. All experiments were performed in replicates. Normalization of gene expression levels was performed using human *β*-actin gene as an endogenous control while healthy controls were used as calibrators. The relative fold change was calculated using 2^−∆∆CT^ method while melting curve analysis was done to assure the presence of specific amplification of the products [17].

### 2.6. Statistical Analyses

Mean, standard deviation and standard error bar were calculated for all regulatory factors in each study subgroup, i.e., endemic control (EC) and non-endemic control (NEC), uncomplicated malaria1 (UC1), uncomplicated malaria 2 (UC2) and severe malaria (SM). Analysis of variance-ANOVA post-hoc (LSD) analysis was performed for regulatory factors/cytokines comparison between all study subgroups. Canonical discriminant function analysis was performed to discriminate study subgroups from each other for all 39 regulatory factors and cytokines. Pearson correlation test was used to assess the correlation between regulatory factors/cytokines and parasitemia versus regulatory factors. Graphical representations for Pearson correlation test and standard error bar graphs at 95% confidence interval (C.I) were executed. Heat-map was generated using ClustVis tool [18]. Statistical Package for Social Sciences version 17 software (SPSS Inc. in Chicago, IL, USA) was used for all statistical analyses and *p*-value < 0.05 was considered statistically significant.

## 3. Results

### 3.1. Clinical and Demographic Characteristics

Demographic and clinical features among different malaria subgroups were compared, as depicted in Table 1. Gender distribution between the groups was markedly discriminated with a higher percentage of males affected by severe malaria. Positive correlation of parasitemia with respiratory distress (*p* < 0.0001) was observed. Age was correlated with BMI (*p* = 0.001) but independent of parasitemia. Elevated pulse rate and respiratory rate (RR) were observed among malaria cases compared to healthy controls. Lower levels of hemoglobin were recorded among severe malaria patients (SM) along with one anemia case (Hb = 6.80 g/dL). Overall, severe malaria (SM) patients were differentiated from the other groups by higher parasitemia levels and respiratory distress.

### 3.2. Altered Expression Levels of Cytokines, Transcription Factors, and Other Signaling Molecules Among Different Malaria Sub-Groups

Cytokines and TF’s level profiling were performed on patient’s blood RNA samples having differential disease burden and were grouped accordingly. A significantly augmented level of IL-1β, IL-10, and TNF-α was found in the SM group compared to the EC and NEC control groups. Interestingly, UC2 groups of these factors showed a marginal decreased level compared to UC1. T helper cell1 (T_H_1) specific master transcription factor T-BET and T helper cell2 (T_H_2) key transcription factor GATA3 levels were found inversely proportional to disease severity among UC1, UC2, and SM malaria sub-groups and were considerably downregulated compared to both of the healthy controls. The consistent decrease in levels was more intense among severe cases than uncomplicated groups with a fold change reduction of 4.74 and 2.01 for T-BET and GATA3, respectively, compared to endemic control. IL-12, as well as receptor IL-12R levels, were also found decreased in all malaria groups compared to healthy subjects. Downregulation of IL-12, coupled with the upregulation of IL-10 with parasitic burden was observed which depicts the host protective mechanism in response to parasitemia load. Evidently, considering the differential expression pattern of the cytokines and other regulatory factors according to disease severity, they were grouped into three major clusters (Table 3).

Cluster 1 factors showed a marked decrease in the expression level with malaria burden. This comprises a total of 23 factors out of 39 cytokines and regulatory factors taken into consideration (Figure 1). On contrary, levels of interferon-gamma (IFN-γ) and Runt-related transcription factor 1 (RUNX1) were found increased during early infection (UC1). However, with malaria progression, they decreased with an intense decrease in SM group. IL-8 and IL-12R, although they showed decreased levels in comparison with the NEC control, they showed their upregulated levels in SM and UC1 groups, respectively, implicating the significance of the endemic over non-endemic healthy controls. Transcription factor STAT4 (signal transducer and activator of transcription 4) was found intensely depressed, similar to STAT6 in the severe group, with a fold change reductions of 2.86 and 1.43, respectively, compared to endemic control.

The second cluster consisted of 15 factors, expression levels of which were found significantly increased with the malaria severity (Figure 2). Not all regulatory factors and cytokine behavior were found idealistic in this cluster as well. For example, the expression level of STAT1 had an upregulated status among all the malaria groups. It was mostly expressed during mild infections (UC1 and UC2 groups) compared to severe cases. Similarly, programmed death-1 (PD-1) which accomplish the immunoregulation during infection along with the GrB, showed elevated expressions during early infections among the UC1 group. However, the levels of GrB together with cMAF and cJUN were found depressed in the UC2 group compared to both healthy control groups. On the other hand, IL-7 was depressed during early infection in both UC1 and UC2 groups. However, its expression was found increased with the malaria severity by a fold change of 1.32 compared to endemic control. It could be explained by upregulation of CD127 receptor following lower expression of T-BET [19] and hence, emerged as a notable factor in host protective mechanism against malaria burden. IL-1β expression was unchanged throughout uncomplicated malaria condition but got upregulated drastically among severe cases by 1.19-fold compared to EC control. Correspondingly, interferon regulatory factor 1 (IRF1) which is an essential regulator of immune responses and inflammation against infection, was found increased but with a constant expression level from the beginning of the infection till it reaches severity. BCL6 (B cell CLL/Lymphoma 6) expression level was also found elevated in the SM group which suggest suppression of macrophage cells proliferation and inhibition of Th1-Th2 cells differentiation in response to malaria severity [20,21]. Surprisingly, the third cluster comprised of single factor NF-κB and was observed discriminated from the other cytokines and regulatory factors by depicting an unchanged or moderately affected expression pattern in all the clinical forms of malaria (Figure 3).

### 3.3. Differential Categorization of Cytokines and Other Regulatory Factors in Malaria Sub-Groups

Canonical discriminant function (CDF) analysis was performed, it separated the different malaria sub-groups notably distinguished from each other. Healthy control groups were separated from the malaria subgroups with an observable distance of endemic control from the non-endemic subjects (Figure 4). A marked distance was more noticeable between SM and EC groups than UC2 and EC groups, which showed the significant parasitic burden on the SM group. Heat-map was generated using log2 fold values for all the cytokines and regulatory factors in all malaria sub-groups using healthy groups as calibrators. Two major gene clusters were observed among cytokines and regulatory factors and grouped into upregulated and downregulated factors (Figure 5). Lower expressions of transcription factor EOMES and PERF by a fold change reduction of 3.90 and 2.20 in SM compared to endemic control were observed. The distressed expression of EOMES may ease natural killer cells, NKT cells, and CD8+ T cells lower proliferation and differentiation which is in accord with malaria burden progression [19]. Similarly, macrophage colony-stimulating factor 1 (CSF1) displayed a significant downregulated level during early infection stage (UC1) and it may facilitate compromised macrophage proliferation and autophagy functionality [22].

Transforming growth factor β (TGF-β) and chemokines like CXCL8 (IL-8), CXCR3, Rantes, and CCR8 were also found downregulated in SM cases by the fold change reduction values of 1.53, 1.25, 1.98, 3.08, and 5.69, respectively, compared to endemic control. Remarkably, immunoregulatory molecules like SOCS1 and SOCS3 were found altered in each malaria sub-groups by the fold change value of 4.04 and 1.88 in UC1 while 4.55 and 3.12 in severe case group, respectively, compared to endemic control. The parasitic burden may craft cytokines and other transcriptional factors differential expression and could be noted for disease manifestation. Fold change analyses for all the factors compared to endemic and non-endemic healthy controls are mentioned in Appendix A.

### 3.4. Correlation Analyses of Cytokines, Transcription Factors and Other Signaling Molecules in Diverse Malaria Subgroups

The correlation analyses of cytokines, TFs, and other parameters were performed to assess the expression consequences of the diverse factors in all malaria groups. IFN-γ was found positively associated with the transcription factors GATA3, EOMES, T-BET, STAT4 and cMAF (Figure 6A). Correlation analysis revealed a positive association of IL-8, IL-10, and cJUN with TNF-α expression, while IL-12, RUNX1, and STAT1 were negatively associated with TNF-α (Figure 6B). Hemoglobin correlation with cytokines and regulatory factors were performed which interestingly, unveiled positive associated with relative mRNA expressions (Ct values) of IL-8, IL-10, GrB, and RUNX3 and an inverse association with STAT1 transcription factor (Figure 6C). Parasitemia was positively correlated with IFN-γ, T-BET, RUNX1, and STAT1 and negatively correlated with NFκB factor (Figure 6D). IL-5, GATA3, RUNX3, and RANTES were inversely associated with IL-1β expression, while NFκB and SOCS3 were positively correlated to IL-1β production (Figure 6E). An interesting and associated regulation of IL-12R was observed with amplified IL-10 production. In addition, overproduction of IL-10 was positively associated with IL-8, GrB, cMAF, and BATF but negatively correlated with STAT1 transcription factor (Figure 6F). IL-10, IL-12R positive correlation was observed with T-BET, GATA3, PERF, RUNX3, STAT4 and cJUN transcription factors (Figure 6G). T_H_2 key transcription factor GATA3 had a positive correlation with T-BET, EOMES, RUNX3, STAT4 and STAT6 factors, while it was found negatively associated with SOCS3 (Figure 6H). PD-1 receptor, which is expressed by several immunoregulatory cells including CD4+ and CD8+ T cells, revealed an increased expression (see Figure 2B) along with decreased T-BET, GATA3 and EOMES expression post-malaria infection (see Figure 1A) which support an inverse association of T-BET with PD-1 [23]. Contrary to this, a positive correlation of PD-1 to T-BET, GATA, and PERFORIN was observed while an inverse correlation was seen with IL-1β, SOCS3 and BCL6 (Figure 6I). It shows more population of PD1+CTLA4+CD4+ T cells over CD4+ T cells during acute malaria [24]. Suppressor of cytokine signaling, SOCS3 was found in a negative association with T-BET and RUNX3 like regulatory factors, while it positively correlated with NFκB, SOCS1 and BCL6 (Figure 6J).

### 3.5. Role of Innate Immunity Factors in Regulation of T_H_1 and T_H_2

A linear regression model was used to conclude the dependencies, as well as the strength of individual singling molecules, among factors of immunoregulatory molecules. Standardized coefficient (Beta) was used to predict the T_H_1 specific cell-mediated immunity factor T-BET and T_H_2 key factor GATA3 as dependent variable and innate immunoregulatory molecules as independent variable. The panel consisted of IL-1β, EOMES, PERFORIN, GRANZY-B, LT alpha, CXCR3, CCR8, ETS1, IRF1, RANTES, BCL6, PD-1 for T-BET and IL-1β, PERFORIN, GRANZY-B, LT alpha, CXCR3, CCR8, ETS1, IRF1, RANTES, BCL6 and PD-1 for GATA3 prediction. Each innate factor was found significantly predicting and hence regulating the cytokines expression and Th1 and Th2 proliferation by influencing master transcription factors T-BET and GATA3 (Table 4).

## 4. Discussion

Immunoregulatory responses and regulations of *P. falciparum* malaria pathogenesis are maintained by the complex coordination of various signaling molecules along with interplay between pro-inflammatory and anti-inflammatory cytokines and other associated molecules (Appendix A). In the present study, the relationship between clinical malaria severity, intricate cytokines, and other signaling molecules multifarious coordination network in subjects from Dhalai district of north-east India was observed. The balance between pro- and anti-inflammatory responses in malaria progression could be best characterized by the expression levels of IL-12 and IL-10 in malaria clinical forms and are equivocally debated too. In contrast to previous reports of higher levels of IL-12 [25,26], IFN-γ [25,27] and lower IL-10 [25,28] expressions in uncomplicated and SM, a lower level of IL-12 in uncomplicated malaria was observed which got further depressed in SM. Further, elevated levels of IL-10 in uncomplicated malaria with a high peak during SM were seen. This data are consistent with the previous findings of low IL-12 [29,30,31] and high IL-10 [27,29,30] expression pattern, depending on malaria severity.

Few reports have also observed a higher IL-12 as well as IL-10 levels following malaria parasitic load [12,26]. Initially, IFN-γ level was found increased and influencing the STAT1-dependent signaling in this study. However, with the parasite burden, both of them got depressed. On contrary to reports of higher circulatory concentrations of TGF-β during malaria infection [12,25], it was found depressed in the present study which is in parallel to previous reports [29,31]. It is crucial for providing protection against malaria severity, significantly in lower concentrations [32,33] given the fact that it was found inversely correlated with parasitemia [34]. Besides, TNF-α and IL-1β levels were also found elevated during SM, similar to previous findings [12,26,35], which reflect the role of malaria pigment or hemozoin in the disease outcome [36]. Monocyte/macrophage arbitrated hemozoin uptake may result in the induction of TNF-α and IL-1β levels [37]. However, TNF-α upregulation was also reported in CD16 dendritic cells along with the induced IL-10 production in response to *P. falciparum* burden [38]. Upregulation of TNF-α could result in augmented expression of Fc receptor on monocytes or modulation of Fc receptor-based signaling pathway mechanism, thus, an increased phagocytic activity against *P. falciparum*. In addition, an augmented TNF-α expression in response to malaria infection could be helpful in parasite clearance and resolution of fever [36]. Hence, following *P. falciparum* infection, the study results obtained insights into a compromised CD4+ T_H_1 activity by downregulation of transcription factor T-BET, master regulator STAT4, LT-α, IFN-γ and receptor IL-12Rβ2. Similarly, a skewed CD4+ T_H_2 condition was observed due to the suppressed transcription factor GATA3, master regulator STAT6, IL-4, IL-5, and IL-13 gene expressions [28].

The reduced level of transcription factor EOMES with lower T-BET expression suggest a compromised innate immunity (lower CD8+Tc, natural killer (NK) cell, and natural killer T (NKT) populations [39]). However, initial augmentation in IFN-γ and STAT1 expression levels in this study depicted their effector functionality during early parasitic infection. Lower levels of major transcription factors like STAT4, ETS1, PERF, and CSF1 (M-CSF) provide validation of compromised innate immunity. Moreover, elevated expression of basic leucine zipper transcription factor ATF like (BATF) factor under the influence of upregulated PD-1, thought to contribute to the suppression of the effector activity of exhausted CD8+T cells while negatively regulating the expression of PERF protein [40,41]. Notably, overexpression of PD-1, observed in this study, authenticates T-cell exhaustion, specifically CD8+T-cell and T-cell dysfunction which is produced in a reduced cytotoxicity and effector functionality of associated cells. Therefore, PD-1 depletion could promote accelerated parasite clearance and T-cell functionality [42,43]. Transcriptional repressor BCL6 promotes the expression of PD-1 and upregulation in both BCL6 and its target PD-1 as observed in the present study and it could facilitate the upregulation of T-follicular helper cell (T_FH_) differentiation [21,44]. A study in Malian children has observed the frequency of circulating PD1^+^CXCR3^−^CXCR5^+^CD4^+^ T_FH_ cells post-malaria infection, which could support B-cells functionality [45]. In favor of this, a lower expression of CXCR3 along with an augmented BCL6 and PD-1 expression in the present study, possibly suggest the increased population of T_FH_ cell following malaria pathology. Majority of the innate immunity regulators like CTLs and NK cells requires the synergy of both PERF and GrB for the caspase-independent killing of an intracellular pathogen [46]. Interestingly, we have witnessed a suppressed PERF expression while GrB mRNA expression was found significantly upregulated in SM. Although PERF is crucial for GrB-mediated initiation of apoptosis, it could not refuse the possibility of augmented GrB uptake by the infected cells via receptor-mediated endocytotic pathway [46,47].

RUNX1 and RUNX3 are well known to play a critical role in T-cell immunity. RUNX1, recognized as a significant hematopoietic regulator in mammals and RUNX3 which acts as an essential neurogenesis controller were also measured through mRNA expression levels [48,49]. RUNX3 is chiefly expressed by T_H_1 committed cells, while T_H_2 requires the synergy of both the RUNX1 and RUNX3 [50]. RUNX3, in association with master transcription regulator TBET enhances T_H_1 differentiation by silencing the expression of IL-4 and promoting IFN-gamma production, this effect could be opposed by GATA3 making a complex with it [51]. In the present study, the downregulated expression pattern of RUNX1 and RUNX3 factors were observed. This reduction was found intensely associated with RUNX1 in the SM group by the fold change reduction of 1.6 and 2.3, respectively, compared to endemic and non-endemic controls. During the early infection (UC1), expression of RUNX1 was found elevated by the fold change value of 1.15 compared to endemic control. Level of RUNX3 was also found decreased in all the malaria sub-groups with a significantly decreased status among UC2 groups compared to both the control subjects.

The present study shows the subordinate expression pattern of chemokine receptors CXCR3 and CCR8 in response to increased malaria burden, it exemplifies the reduced trafficking of T_H_1, CD8+T cells, NK cells, monocytes and T_H_2 cells within lymphoid organ or in the peripheral tissues [52,53]. However, a study in the murine model revealed an upregulated CXCR3 expression in NK cell and T-cell and its association with lymphocyte trafficking during cerebral malaria [54] and mice deficient in CXCR3 was found efficiently protected from cerebral malaria [55]. The reduced level of RANTES (CCL5) was also observed which seem to be associated with increased disease severity and was similar to the previous finding [56]. Additionally, reduced circulatory level of RANTES was found correlated with severe malarial anemia in children and associated with suppression of erythropoiesis and parasite-induced thrombocytopenia [56,57]. On the other hand, contrary to the reports of the higher circulating level of IL-8 (CXCL8) in response to *P. falciparum* malaria [58,59], depressed expression of IL-8 was observed frequently during early infection. Notably, the expression of IL-8 was found increased in correlation with the parasite densities during severe condition compared to uncomplicated cases.

The study illustrates the differential transcriptional regulations and suppression of innate as well as cell-mediated immunity in the diverse clinical subgroups of *P. falciparum* malaria infection. Various signaling molecules collectively produce complex signaling cascade which proclaims an active role in the malaria progression with their differential expression levels in accord with disease burden. These could be helpful as indicative biomarkers of malaria as well as for monitoring the disease severity. Enhanced TNF-α, IL-1β, IL-10, and IL-7 production and lower TGF-β concentrations, however, may together provide some protection against the parasitic deleterious effects. It was a model study with a modest sample size aimed to identify the biomarkers responsible for the progression from uncomplicated to severe malaria. A study in a large sample size could be more beneficial in predicting the malaria diagnosis using measurement of the panel’s one or more cytokines, transcription factors, or other signaling molecules.

## Figures and Tables

**Figure 1 biomolecules-09-00600-f001:**
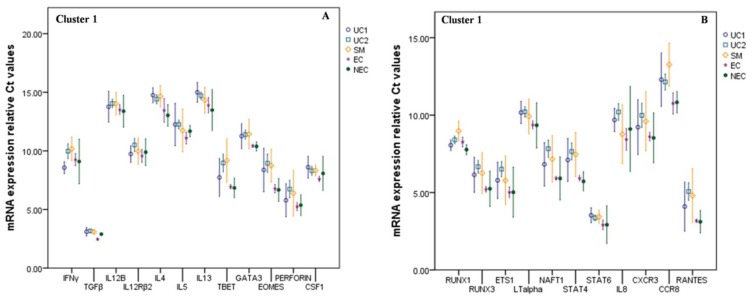
Relative mRNA expression profiling of cytokines and other regulatory factors showing differential disease burden among malaria subgroups are presented as Ct values. Expressions of these signaling molecules were found decreased with the *P. falciparum* malaria burden and were grouped as cluster 1. Major T_H_1, T_H_2 factors, and players of innate immunity were found depressed (**A**) IFNG displayed gene upregulation during early malaria progressions. Master transcription factor of T_H_1 (T-BET), T_H_2 (GATA3) as well as CD8+ Tc (EOMES) cells got depressed as malaria progress. Expressions of PERFORIN and macrophage cells specific transcription factor CSF1 were found downregulated. (**B**) Grouped as innate immunity regulatory molecules, RUNX1 and RUNX3 were found depressed. However, RUNX1 displayed gene upregulation during early malaria progressions compared to healthy endemic control subjects. Chemokines and chemokines receptors, including STAT4, STAT6, and ETS1 levels were found altered. Uncomplicated malaria1 (UC1 = parasitemia < 25,000/μL), uncomplicated malaria2 (UC2 = parasitemia > 25,000/μL), severe malaria (SM), endemic control (EC), non-endemic control (NEC). *β*-actin was used as an endogenous control. Mean and standard error bars were calculated at a 95% confidence interval (CI). Gene expression decreases as Ct value increases and vice versa.

**Figure 2 biomolecules-09-00600-f002:**
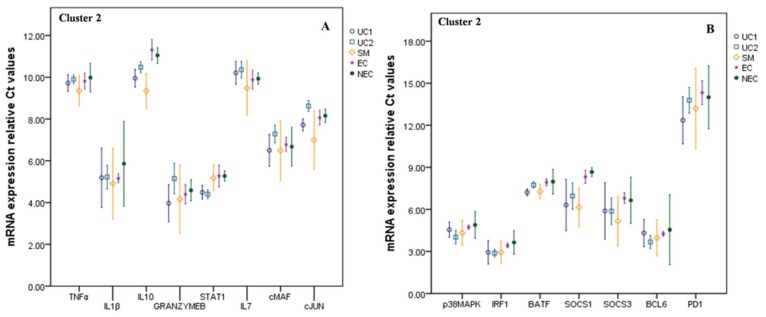
Expressions of these signaling molecules got upregulated following malaria progression and were grouped as Cluster 2. (**A**) Cytokines like TNF-α, IL10, and IL1β levels were upregulated with an increase in malaria burden. STAT1 expression was moderately upregulated during early infections. Despite downregulated perforin expression, GrB mRNA expression was elevated in severe malaria. Expression levels of IL7 and cJUN were low during the early stages (UC1 and UC2) but were considerably upregulated in severe malaria (SM). (**B**) p38MAPK mRNA expressions were upregulated. Negative regulators of cytokine signaling like SOCS1 and SOCS3 found elevated following malaria burden. T cell exhaustion indicator, PD1 was also got upregulated following malaria burden. Uncomplicated malaria1 (UC1 = parasitemia < 25,000/μL), uncomplicated malaria2 (UC2 = parasitemia > 25,000/μL), severe malaria (SM), endemic control (EC), non-endemic control (NEC). *β*-actin was used as an endogenous control. Mean and standard error bars were calculated at a 95% confidence interval (CI). Gene expression decreases as Ct value increases and vice versa.

**Figure 3 biomolecules-09-00600-f003:**
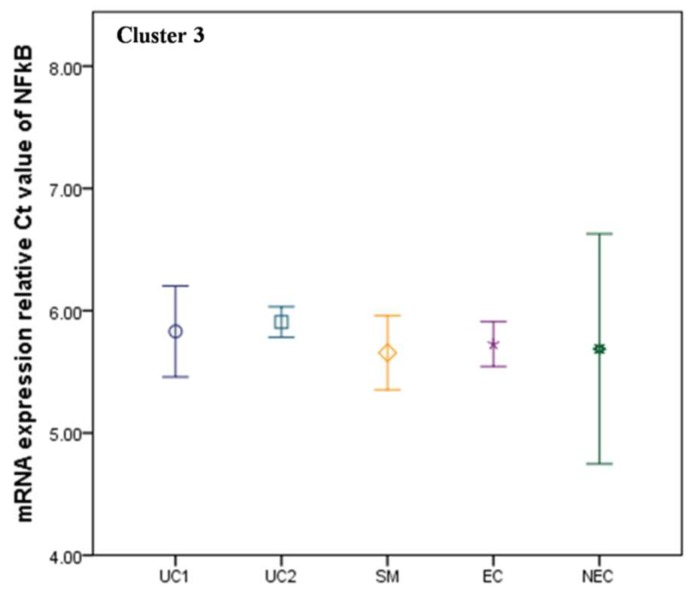
NF-κB expression was not much influenced following malaria burden in any of the clinical subgroups and was grouped in Cluster 3 separately. Uncomplicated malaria1 (UC1 = parasitemia < 25,000/μL), uncomplicated malaria2 (UC2 = parasitemia > 25,000/μL), severe malaria (SM), endemic control (EC), non-endemic control (NEC). *β*-actin was used as an endogenous control. Mean and standard error bars were calculated at 95% confidence interval (CI). Gene expression decreases as Ct value increases and vice versa.

**Figure 4 biomolecules-09-00600-f004:**
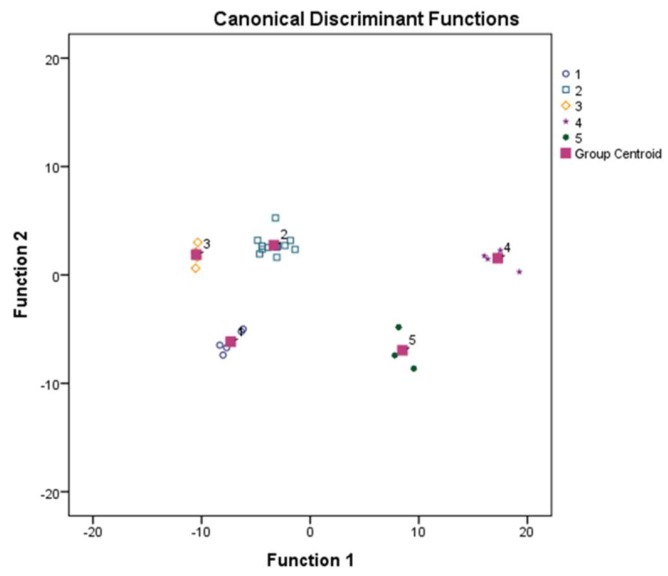
Canonical discriminant function (CDF) illustrates the characteristics of the different clinical subgroups considering regulatory factors/cytokine expression profiles in *Plasmodium falciparum-*infected patients and healthy control subjects. Function 1 and 2 resulting from CDF, discriminating the EC and NEC from uncomplicated malaria1 (UC1), uncomplicated malaria2 (UC2) and severe malaria (SM) groups according to their relative mRNA expressions (Ct values). Annotation indicates the factors (cytokines and TF’s) that turned out to be relevant for each type of discrimination. CDF discrimination of the clinical subgroups was analyzed using all 39 factors considered for the study. 1 = UC1 (parasitemia < 25,000/μL), 2 = UC2 (parasitemia > 25,000/μL), 3 = SM, 4 = EC and 5 = NEC.

**Figure 5 biomolecules-09-00600-f005:**
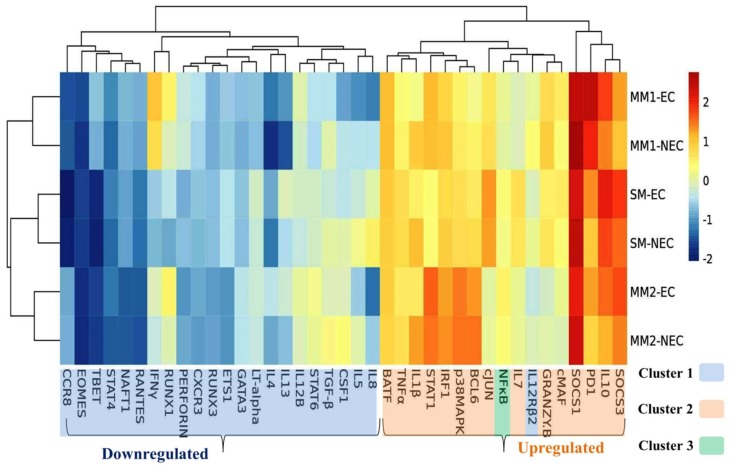
Heat-map illustrates comparative fold change of each group per gene expression profile. Each cell of the 2D plot denotes log2 fold value of a single factor in one group with standardized levels represented by color as per color scale on the right. Fold change was calculated by 2^−ΔΔCt^ method considering *β*-actin as an endogenous control. Top horizontal and left vertical dendrogram indicates average Euclidean clustering of genes taken into panel and malaria clinical subgroups, respectively. UC1 was marked and sub-grouped from the UC2 and SM group. The figure shows a consequence of higher parasitic burden displayed by the latter two groups despite other reasonable associated causes. UC1-EC, uncomplicated malaria group (parasitemia < 25,000/μL) fold change compared to endemic control; UC1-NEC, uncomplicated malaria group (parasitemia < 25,000/μL) fold change compared to non-endemic control; UC2-EC, uncomplicated malaria group (parasitemia > 25,000/μL) fold change compared to endemic control; UC2-NEC, uncomplicated malaria group (parasitemia > 25,000/μL) fold change compared to non-endemic control; SM-EC, severe malaria group fold change compared to endemic control; SM-NEC, severe malaria group fold change compared to non-endemic control.

**Figure 6 biomolecules-09-00600-f006:**
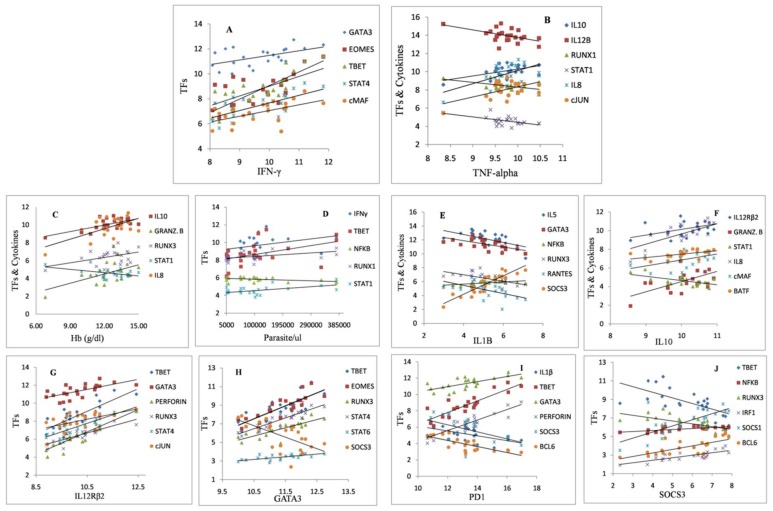
Pearson correlation statistical tests were performed to assess the correlation between cytokines, transcription factors, parasitemia and hemoglobin in all malaria sub-groups (UC1+ UC2+ SM). (**A**) *r* = 0.512, 0.656, 0.831, 0.658, 0.561, *p* = 0.015, 0.001, 0.000, 0.001, 0.007 (**B**) *r* = 0.519, −0.509, −0.454, −0.551, 0.571, 0.542, *p* = 0.013, 0.016, 0.034, 0.008, 0.005, 0.009 (**C**) *r* =0.669, 0.504, 0.442, −0.455, 0.617, *p* = 0.001, 0.017, 0.039, 0.033, 0.002 (**D**) *r* = 0.442, 0.438, −0.443, 0.471, 0.539, *p* = 0.039, 0.041, 0.039, 0.027, 0.010 (**E**) *r* = −0.561, −0.619, 0.651, −0.571, −0.521, 0.905, *p* = 0.007, 0.002, 0.001, 0.006, 0.013, 0.000 (**F**) *r* =0.454, 0.588, −0.619, 0.632, 0.482, 0.581, *p* = 0.034, 0.004, 0.002, 0.002, 0.023, 0.005 (**G**) *r* = 0.804, 0.651, 0.856, 0.843, 0.853, 0.559, *p =* 0.000, 0.001, 0.000, 0.000, 0.000, 0.007 (**H**) *r* = 0.768, 0.840, 0.804, 0.896, 0.676, −0.622, *p =* 0.000, 0.000, 0.000, 0.000, 0.001, 0.002 (**I**) *r* = −0.487, 0.869, 0.696, 0.850, −0.484, −0.756, *p* = 0.022, 0.000, 0.000, 0.000, 0.023, 0.000 (**J**) *r* = −0.662, 0.646, −0.555, 0.777, 0.699, 0.694, *p =* 0.001, 0.001, 0.007, 0.000, 0.000, 0.000.

**Table 1 biomolecules-09-00600-t001:** Age, gender distribution, parasitemia level and other demographic details among different malaria clinical groups.

	UC1 (*n* = 05)	UC2 (*n* = 12)	SM (*n* = 05)	EC (*n* = 06)	NEC (*n* = 03)
**Age**	16.8 ± 4.96 *	26.58 ± 10.70	22.8 ± 11.78	30 ± 8.32 *	34 ± 5.29
**Gender F [%]**	4 (80)	2 (16.7)	1(20)	3 (50)	1(33.3)
**BMI**	17.92 ± 1.44 *	18.54 ± 2.53 **	19.18 ± 2.24 ^&^	22.80 ± 3.60 *^,^ **^,^ ^&^	24.96 ± 2.38
**Parasite/μL**	9104 ± 3242.42	90184.75 ± 28871.36	289859.6 ± 98092.5	NA	NA
**Systolic B.P**	116.2 ± 10.92	108.16 ± 18.38 **	100.6 ± 14.10 ^&^	133.33 ± 8.62 **^,^ ^&^	116 ± 8.89
**Diastolic B.P**	79.4 ± 9.42 ^#^	70.33 ± 9.54 **	68.2 ± 4.87 ^#, &^	86.66 ± 4.13 **^,^ ^&^	76.33 ± 5.69
**Pulse**	110 ± 16.19 *	92.41 ± 18.48	97 ± 18.17	82.5 ± 11.15 *	84 ± 2.0
**RR**	22.6 ± 1.52 ^#, $^	25.83 ± 2.48 **^,^ ^##, $^	36.4 ± 1.14 ^#, ##, &^	20.83 ± 1.83 **^,^ ^&^	19 ± 1.0
**Hb [g/dL]**	12.34 ± 1.05	13.09 ± 0.86 ^##^	11.2 ± 3.17 ^##^	12.60 ± 1.69	14.53 ± 0.32
**HCT [%]**	37.6 ± 3.21	39.08 ± 3.45	34.8 ± 10.99	39.83 ± 5.34	43.33 ± 1.53

*p*-values were calculated using ANOVA post-hoc (LSD) analysis. Parameters were represented as mean and standard deviation (±). Note: Endemic control (EC), non-endemic control (NEC), uncomplicated malaria1 (UC1 = parasitemia < 25,000/μL), uncomplicated malaria2 (UC2 = parasitemia > 25,000/μL) and severe malaria (SM).^*^ = UC1 vs. EC, *p*-value < 0.05; ^**^= UC2 vs. EC, *p*-value < 0.05; ^#^ = UC1 vs. SM, *p*-value < 0.05; ^##^ = UC2 vs. SM, *p*-value < 0.05; ^$^ = UC1 vs. UC2, *p*-value < 0.05; ^&^ = SM vs. EC, *p*-value < 0.05.

**Table 2 biomolecules-09-00600-t002:** List of primer sequences designed for SYBR green-based quantitative real-time polymerase chain reaction.

S.N	Genes (*Homo Sapiens*)	HGNC Gene ID	Forward Primer	Reverse Primer	Product Size (bp)	Tm	DDBJ Accession Numbers	NCBI Chromosome Location
1	IFN-γ (IFNG)	HGNC:5438	GCAGCCAACCTAAGCAAGAT	CAAACCGGCAGTAACTGGAT	103	60	LC461674	12q15
2	TNF-α (TNF-alpha)	HGNC:11892	GCCCGACTATCTCGACTTTG	GGTTGAGGGTGTCTGAAGGA	141	60	LC461675	6p21.33
3	TGF-β1 (TGFB1)	HGNC:11766	CCCTGGACACCAACTATTGC	CAGAAGTTGGCATGGTAGCC	130	60	LC461676	19q13.2
4	β-ACTIN (ACTB)	HGNC:132	TCGTGCGTGACATTAAGGAG	GTCAGGCAGCTCGTAGCTCT	110	60	LC461677	7p22.1
5	IL-1β (IL1B)	HGNC:5992	GGCGGCCAGGATATAACT	CCCTAGGGATTGAGTCCACA	100	60	LC461678	2q14.1
6	IL-4	HGNC:6014	GGCTTGAATTCCTGTCCTGT	ATGATCGTCTTTAGCCTTTC	77	60	LC461679	5q31.1
7	IL-5	HGNC:6016	AGGGCCAAGAAAGAGTCAGG	TGCCTGGAGGAAAATACTTC	153	60	LC461680	5q31.1
8	IL-7	HGNC:6023	TTCCTCTGGTCCTCATCCAG	ATCCGCCAGCAGTGTACTTT	140	60	LC461681	8q21.13
9	IL-8 (CXCL8)	HGNC:6025	CTAGGACAAGAGCCAGGAAGAA	AACTGCACCTTCACACAGAGC	128	60	LC461682	4q13.3
10	IL-10	HGNC:5962	TTGGGGCTTCCTAACTGCTAC	AGTGGTTGGGGAATGAGGTTAG	118	62	LC461683	1q32.1
11	IL-12B	HGNC:5970	ATTGTGCCACTGCATACCAG	AGGACTGCCATGGAAGCTAA	101	62	LC461684	5q33.3
12	IL-12Rβ2	HGNC:5972	ACTGGAGCCTCAGCACATCT	AGCCTCACCACTCAGAGCAT	138	60	LC461685	1p31.3
13	IL-13	HGNC:5973	GCCAAGGGTTCAGAGACTCA	GACCCCAGTGAGGTAGCAGA	102	60	LC461686	5q31.1
14	RUNX1	HGNC:10471	GGGAACTGTCAAGCTGGTGT	CTGTGTACCGTGGACTGTGGA	126	58	LC461687	21q22.12
15	RUNX3	HGNC:10473	TGAGAGGTGGGGAGTACTGG	GGCAAGACTTCACCTCGGAA	102	60	LC461688	1p36.11
16	IRF1	HGNC:6116	GAAGAACATGGATGCCACCT	TCTCTGCACCATATCCACCA	156	60	LC461689	5q31.1
17	T-BET (TBX21)	HGNC:11599	GGAAACGGATGAAGGACTGA	ATCCTTCTTGAGCCCCACTT	89	58	LC461690	17q21.32
18	GATA3	HGNC:4172	GAGGGTAGCAGTGTATGAGCT	CACTAACACAGAACACGACAGG	112	58	LC461691	10p14
19	STAT1	HGNC:11362	ACAAAGTCATGGCTGCTGAG	AAGTTCCATTGGCTCTGGTG	128	60	LC461692	2q32.2
20	STAT4	HGNC:11365	CAACCAACGATTCCCAGAAC	TCTGCCAGCATATGGAGTTG	142	58	LC461693	2q32.2-q32.3
21	STAT6	HGNC:11368	AACATCCAGCCATTCTCTGC	TTGGGCTTCTTGGGATAGAG	101	58	LC461694	12q13.3
22	NF-KB1 (NFKB1)	HGNC:7794	CTGGAAGCACGAATGACAGA	TGAGGTCCATCTCCTTGGTC	172	60	LC461695	4q24
23	EOMES	HGNC:3372	CCACTGCCCACTACAATGTG	CTCATCCAGTGGGAACCAGT	166	60	LC461696	3p24.1
24	GrB (GZMB)	HGNC:4709	CCAGGGCATTGTCTCCTATG	ATTACAGCGGGGGCTTAGTT	138	60	LC461697	14q12
25	PERFORIN (PRF1)	HGNC:9360	CATGTAACCAGGGCCAAAGT	GGCTTAGGAGTCACGTCCAG	104	60	LC461698	10q22.1
26	CSF1	HGNC:2432	TAAGAGACCCTGCCCTACCTG	CAAGTTCACTGCCCTTCCCTA	127	58	LC461699	1p13.3
27	LT-ALPHA (LTA)	HGNC:6709	CCTGATGTCTGTCTGGCTGA	TGCTCTTCCTCTGTGTGTGG	113	60	LC461700	6p21.33
28	CXCR3	HGNC:4540	AGCTTTGACCGCTACCTGAA	GCCGACAGGAAGATGAAGTC	140	60	LC461701	Xq13.1
29	CCR8	HGNC:1609	CCCTGTGATGCGGAACTTAT	CAGACCACAAGGACCAGGAT	119	60	LC461702	3p22.1
30	cMAF	HGNC:6776	CTTTGCTCTCTGCCTCGTCT	CGCTCTCTACCTCTGTGCAA	141	60	LC461703	16q23.2
31	NFAT1 (NFATC2)	HGNC:7776	CTGGAGGTGGGTTTCTACCA	AGGGGCAGAAGGGATCTTTA	134	60	LC461704	20q13.2
32	cJUN(AP1)	HGNC:6204	CACGTGAAGTGACGGACTGT	CAGGGTCATGCTCTGTTTCA	143	60	LC461705	1p32.1
33	p38 MAPK (MAPK14)	HGNC:6876	TGCACATGCCTACTTTGCTC	AGGTCAGGCTTTTCCACTCA	116	60	LC461706	6p21.31
34	SOCS1	HGNC:19383	AGACCCCTTCTCACCTCTTGA	TAGGAGGTGCGAGTTCAGGT	117	60	LC461707	16p13.13
35	SOCS3	HGNC:19391	GAGACGGGACATCTTTCACCT	CAGGCTGAGTATGTGGCTTTC	152	60	LC461708	17q25.3
36	BATF	HGNC:958	GGAGTGAACACGGGAACTGT	CCATGGGACTTGAGCATCTT	148	60	LC461709	14q24.3
37	BCL6	HGNC:1001	CAGCCACAAGACCGTCCATAC	CGAGTGTGGGTTTTCAGGTTG	96	60	LC461710	3q27.3
38	ETS1	HGNC:3488	TGGTCTAGCTGGGTGAAACC	CCAGAATGGAGAAGGGAACA	102	60	LC461711	11q24.3
39	PD-1 (PDCD1)	HGNC:8760	CCTGCAGGCCTAGAGAAGTTT	GGGCATGTGTAAAGGTGGAG	91	60	LC461712	2q37.3
40	RANTES (CCL5)	HGNC:10632	TCTGTGACCAGGAAGGAAGTC	GTTTGCCAGTAAGCTCCTGTG	108	60	LC461713	17q12

**Table 3 biomolecules-09-00600-t003:** Differential clustering of TF’s and cytokines based on their similar expression order.

Cluster 1		Cluster 2		Cluster 3
IFNγ	Runx3	TNFα	p38	NFκB
TGF-β	Stat4	IL10	BATF	
IL12	Stat6	IL1β	BCL6	
GATA3	IL12Rβ2	Granzyme B	IRF1	
Tbet	CXCR3	Stat1	PD-1	
Eomes	CCR8	IL7		
IL4	LTα	cMAF		
IL5	NFAT	cJUN		
IL13	ETS1	Socs1		
CSF1	IL8	Socs3		
Perforin	Rantes			
Runx1				

**Note.** Cluster 1= Factors with decreased expression following the disease severity, Cluster 2 = Factors with increased expression following the disease severity, Cluster 3 = Factors with moderate or unchanged expression following the disease severity.

**Table 4 biomolecules-09-00600-t004:** Linear regression predictive analysis of immunoregulatory factors.

Factors (All Malaria Cases)	TBET **		Factors (All Malaria Cases)	GATA3 ***	
	Standardized Coefficient (Beta)	*p*-Value *		Standardized Coefficient (Beta)	*p*-Value *
IL-1β	−0.645	0.001	IL-1β	−0.619	0.002
EOMES	0.909	0.000	PERFORIN	0.639	0.001
PERFORIN	0.882	0.000	GRANZY-B	0.418	0.053
GRANZ-B	0.639	0.001	LT alpha	0.740	0.000
LT alpha	0.610	0.003	CXCR3	0.769	0.000
CXCR3	0.864	0.000	CCR8	0.474	0.026
CCR8	0.435	0.043	ETS1	0.859	0.000
ETS1	0.810	0.000	IRF1	−0.503	0.017
IRF1	−0.555	0.007	RANTES	0.747	0.000
RANTES	0.868	0.000	BCL6	−0.549	0.008
BCL6	−0.738	0.000	PD-1	0.696	0.000
PD-1	0.869	0.000			

* *p*-values were calculated using a linear regression model analysis. ** Dependent variable: T-BET, *** dependent variable: GATA3.

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
