# Peer review of "Transcriptional Modulation of the Host Immunity Mediated by Cytokines and Transcriptional Factors in *Plasmodium falciparum*-Infected Patients of North-East India"

_biomolecules, 2019, doi:10.3390/biom9100600_

Round 1

Reviewer 1 Report

Dear authors,

It is not clear to the reader what is novel about your paper. You applied classical methodology; apparently, the majority of the presented observations have already been published. It is unclear whether there is something new. In my opinion, you should clearly indicate what observations are new and have not been published yet.

I also respectfully request that you improve your citations. For example, while discussing your results concerning GATA3, you did not cite the paper “Lower expression of GATA3 and T‐bet correlates with downregulated IL‐10 in severe falciparum malaria” (although you mentioned this paper in the case of IL-10).

Best wishes,

Author Response

Point 1: It is not clear to the reader what is novel about your paper. You applied classical methodology; apparently, the majority of the presented observations have already been published. It is unclear whether there is something new. In my opinion, you should clearly indicate what observations are new and have not been published yet.

Response 1: The authors are thankful to the reviewer for reviewing the manuscript and welcome his concern. The authors agree with the reviewer opinion about the applied classical methodology in the study. The study illustrates the differential transcriptional regulations and suppression of innate as well as cell-mediated immunity in the diverse clinical subgroups of P. falciparum malaria infection and is a new approach towards study. Authors presented the correlation of differential parasitic burden with respect to the malaria infection as well as its signaling responses and modulations in the host. The signaling factors were clustered in three categories based on malaria parasite role in the host. The study demonstrates the role of parasitic burden in the host immune system modulation and hence in disease outcome.

Various signaling molecules collectively produce complex signaling cascade which proclaims an active role in the malaria progression with their differential expression levels in accord with disease burden. These could be helpful as an indicative biomarker of malaria as well as for monitoring the disease severity. It was a model study with modest sample size aimed to identify the biomarkers responsible for the progression of uncomplicated to severe malaria. A study in the large sample size could be more beneficial in predicting the malaria diagnosis using the measurement of the panel’s one or more cytokines, transcription factors, or other signaling molecules.

Point 2: I also respectfully request that you improve your citations. For example, while discussing your results concerning GATA3, you did not cite the paper “Lower expression of GATA3 and T‐bet correlates with downregulated IL‐10 in severe falciparum malaria” (although you mentioned this paper in the case of IL-10).

Response 2: As per the suggestion of the reviewer the reference has been added at respective place.

 Page No - 14   Line no - 383

Reviewer 2 Report

Malaria caused by Plasmodium falciparum is the most prevalent form of the disease in India. In this paper, the authors correlate the occurrence of P. falciparum malaria and the levels of different immune markers. The authors reported extensive immunomodulation and suppression of chemokines, chemokine receptors, and some transcription factors.

Overall the experimental design is appropriated, the paper is informative, and the multifaceted analysis is of great value for the understanding of the outcome of the disease.

There is some general point that I think the authors should address: 

1- The authors described in the material and methods, line 92, that the patients were diagnosed with P. falciparum malaria through light microscopy. Though it is estimated that P. falciparum causes 90% of the disease in Tripua, P. vivax infection is still prevalent, and it is estimated to account for 10% of the cases. Moreover, the occurrence of mixed cases should not be ruled out. Considering these premises, the microscopy diagnose is not the most reliable way for the diagnostic of malaria at the species level. How can the authors rule out the effect of other species on the immune response genes considered here?

2- One point that would be worth to consider for discussion is whether there the patients were previously infected with malaria. What is the impact of multiple infections on the clustering? 

Minor points:

1- The heat map (figure 5) can be improved. For clarity, it will be useful to add to the top of the graph, for each of the three clusters, a color bar, so that one can quickly identify them. 

Author Response

Point 1: The authors described in the material and methods, line 92, that the patients were diagnosed with P. falciparum malaria through light microscopy. Though it is estimated that P. falciparum causes 90% of the disease in Tripura, P. vivax infection is still prevalent, and it is estimated to account for 10% of the cases. Moreover, the occurrence of mixed cases should not be ruled out. Considering these premises, the microscopy diagnose is not the most reliable way for the diagnostic of malaria at the species level. How can the authors rule out the effect of other species on the immune response genes considered here?

Response 1: Authors welcome the reviewer’s suggestions and are thankful to raise this important issue, the samples were screened with malaria rapid diagnostic test (RDT) (Pf/Pv) along with microscopy. For further validation diagnostic PCR was performed for the confirmation of mono infection of P. falciparum.

Point 2: One point that would be worth to consider for discussion is whether there the patients were previously infected with malaria. What is the impact of multiple infections on the clustering? 

Response 2: History of all the participants was obtained. None of the study participants had a history of other complications before malaria infection. The authors welcome the valuable query on the impact of multiple infections on the clustering. However, the present study was aimed to identify the immunological biomarkers of complications and malaria severity in a small group on a pilot basis. It was conducted in order to evaluate the feasibility and to improve upon the study design prior to performance of a full-scale research project. On the basis of the results of this pilot study, further research is being planned. 

Minor points:

1- The heat map (figure 5) can be improved. For clarity, it will be useful to add to the top of the graph, for each of the three clusters, a color bar, so that one can quickly identify them. 

As per the suggestion of the reviewer, the clustering is rearranged in the heat map. The three clusters are shown in three different colors along with the colour bar provided.

The figure 5 is updated in the manuscript in Page no. 12, Line no- 279.
